# Hypothetical Model of How a Clinical Remount Procedure Benefits Patients with Existing Dentures: A Narrative Literature Review

**DOI:** 10.3390/healthcare10061067

**Published:** 2022-06-09

**Authors:** Chi-Hsiang Cheng, Ikiru Atsuta, Kiyoshi Koyano, Yasunori Ayukawa

**Affiliations:** 1Section of Implant and Rehabilitative Dentistry, Division of Oral Rehabilitation, Faculty of Dental Science, Kyushu University, Fukuoka 8128582, Japan; teikeishoudds@gmail.com (C.-H.C.); koyano@dent.kyushu-u.ac.jp (K.K.); ayukawa@dent.kyushu-u.ac.jp (Y.A.); 2Division of Advanced Dental Devices and Therapeutics, Faculty of Dental Science, Kyushu University, Fukuoka 8128582, Japan

**Keywords:** clinical remount, complete denture, malocclusion

## Abstract

The clinical remount procedure, which involves remounting the dentures on an articulator with interocclusal records, can effectively reduce occlusal discrepancies. This procedure can be applied not only to new dentures but also to those already in service; however, research in this field is still scarce. This narrative review aims to establish a hypothetical mechanism and possible indications and contraindications for this technique as a basis for further research. Current studies have revealed a high prevalence of malocclusion in delivered dentures. Performing a clinical remount on these existing dentures would enhance the oral function of the denture wearer and would enable effective and accurate correction of the accumulated errors in the jaw relationship in a stable working environment. This technique should be performed if a patient has poor masticatory function or occlusion-related complaints. However, performing a clinical remount on dentures with an excessive anterior–posterior discrepancy between the centric relation and the maximal intercuspal position or on dentures with extremely low occlusal vertical dimension, is considered less effective. The clinical remount procedure remains an essential skill both for fabricating quality dentures and maintaining those already in service.

## 1. Introduction

Occlusal harmony and an accurate, reproducible jaw relationship are necessary for complete dentures to function efficiently [1] (p. 394) [2]. Occlusal harmony can be achieved using various intraoral and extraoral methods.

The occlusion of dentures can be corrected in patients’ mouths directly using articulating paper, occlusion wax or central-bearing devices [1] (p. 395). The intraoral method is intuitive and convenient, and procedures can be interrupted and continued according to the appointment time. However, it is clear that the results of the adjustment may be jeopardized by displacement of the denture supporting tissues, interference of saliva, inconsistent markings and confined vision [3,4,5,6,7].

The extraoral method, or adjusting the occlusion of complete dentures on an articulator, is often referred to as the remount procedure. The clinical remount procedure is to remount the dentures on an articulator with interocclusal records made in the patient’s mouth [1] (p. 397). Establishing the occlusion and articulation of dentures outside the patient’s mouth overcomes the flaws of the intraoral method. Although this procedure is considered complex and time consuming [6,8,9], previous research has established that a clinical remount can efficiently remove occlusal interference, enhance comfort and masticatory performance and reduce the incidence of soreness and tissue irritation [6,10,11,12,13,14,15,16,17,18,19].

However, most research about clinical remounting has focused on newly fabricated dentures. If clinical remounting is effective as a method of occlusal adjustment in new dentures, it should also provide some benefits for existing dentures. Although previous expert opinions suggested that periodical clinical remounts should be implemented in the maintenance phase of complete denture treatment [20] (p. 113) [21] (p. 227), there is an absence of sufficient research and treatment guidelines [22] or any form of review.

Herein, this narrative review aims to evaluate the available evidence about performing clinical remounts on existing dentures. A hypothetical treatment mechanism, possible indications and contraindications will also be established as a basis for future research.

## 2. Materials and Methods

Studies with the following properties were considered eligible: written in English or Japanese, original research about the observational findings or treatment results of performing clinical remount procedures on existing removable dentures. Case reports, technical papers, expert opinions and abstracts without a complete article were excluded. No restrictions were applied to the date of publication.

The electrical databases PubMed and Ichushi (a Japanese online medical database operated by Non-Profit Organization Japan Medical Abstracts Society) were searched in January 2022 and updated in May 2022. Keywords “clinical remount” were applied for PubMed and “リマウント (Rimaunto)” for Ichushi. The latter keyword is a loanword in Japanese Katakana, meaning “remount” (Table 1). After the initial screening of titles and abstracts, the reference lists of included articles were inspected for additional studies that might meet the inclusion criteria.

Screening and data extraction were completed by the author (C.C.). Author, research objectives, mounting procedures, methods of evaluation, the profile of participants and brief discussion of the included studies were listed.

## 3. Results—Brief Review of Current Available References

### 3.1. Inclusion Results

The inclusion process was illustrated (Figure 1). After the initial screening of titles and abstracts, 16 articles were included for full-test inspection. Four case reports [23,24,25,26], five expert opinions [13,15,20,21,27] and four technical papers [28,29,30,31] were excluded (Table 2). Three original studies, two written in English [5,32] and one in Japanese [33], were included, reviewed and listed eventually (Table 3).

### 3.2. Review of Current Available Research

Schierano et al. performed a thickness discrimination test with metal foil to evaluate the oral function of participants [32]. Two groups of 12 edentulous patients were previously rehabilitated with either bimaxillary traditional complete dentures (CD) or a conventional maxillary denture combined with a mandibular implant-supported overdenture (IOD). The average duration since rehabilitation was 4 ± 0.42 years for the CD group and 4.42 ± 0.79 years for the IOD group. The prostheses were mounted on an articulator with the aid of a dynamic facebow and a centric bearing device. Selective grinding was then performed if necessary, but the acquired occlusal scheme after selective grinding was not mentioned. A thickness discrimination test was performed before and 2 months after selective grinding, which was needed in 23 out of 24 patients. The mean thickness discrimination ability of the CD group decreased from 110 to 90.5 μm after grinding, and the measurements in the IOD group decreased from 64 to 37.67 μm, indicating that the participants had more sensitive thickness perception after occlusal adjustment. According to Schierano et al., the occlusal alterations affected nearly all the patients and were not correlated with the time since rehabilitation.

Kawahara et al. gathered 70 participants with their existing complete dentures and measured the masticatory function with a standardized gummy testing kit (Glucosensor GS-I, GC, Tokyo, Japan) before and immediately after the clinical remount procedure [33]. The mean age of the participants was 75.8 years, but the duration since rehabilitation was not mentioned. The existing complete dentures were mounted on a semi-adjustable articulator without facebow transfer. Bilateral balanced articulation was achieved after adjustment. The average reading of the testing kit was increased from 135.1 to 184.4 mg/dL, indicating the increment in the masticatory function. Further analysis showed that the participants with lower masticatory function would benefit most after the clinical remount procedure.

Atashrazm et al., though, did not mention the specific treatment results after performing clinical remounts on existing dentures, providing the prevalence of occlusal disharmony in inserted dentures [5]. In total, 107 participants were gathered, and new bimaxillary complete dentures were fabricated and delivered. All participants were recalled within 30 days after delivery, and their dentures were remounted to an articulator with facebow transfer. The occlusion of the mounted dentures was then inspected with 60 μm articulating paper by tactile sensation. The occlusal disharmony was defined as an absence of bilateral simultaneous contact of posterior teeth, with at least three contact points for each side. Information including age, gender, ridge relationships, the occlusal scheme of dentures and whether the dentures had clinical remount performed or not at the insertion appointment was recorded as well. Among 107 participants, 74 of them had received clinical remount, while 33 of them had not. A total of 31 participants were recorded with occlusal disharmony, and 6 of them had received clinical remount, while the rest of the 25 participants had not. The prevalence of occlusal disharmony was highly relevant to performing clinical remount at the insertion appointment instead of age, gender, ridge relationships or the occlusal scheme applied, with an odds ratio > 35.

### 3.3. Brief Discussion

All the aforementioned studies evaluated mounted dentures at the centric relation (CR). Atashrazm found some new, carefully adjusted dentures that showed occlusal disharmony after a short period of usage, not to mention those without clinical remount procedure at the insertion appointment. These findings were in line with the studies published by Neil [20] and Utz [14]. Schierano and Kawahara also reported that most of their participants suffered from discrepancies of occlusion. After performing the clinical remount procedure again, the oral function of the participants was enhanced.

In 2019, Verhaeghe et al. published a systematic review about clinical remount [19]. Four randomized controlled trials (RCT) were included, and all of these studies focused on the clinical effects of performing clinical remount at the delivery of new complete dentures. By doing so, patients’ discomfort and the number of required recalls were reduced.

It is widely suggested that the delivered dentures can be remounted back on articulators and re-balanced if necessary [13,14,15,20,21,34,35] (p. 113 of Ref [20]) (p. 227 for Ref [21]) (p. 279 of Ref [34]); however, the treatment efficiency is seldom discussed. Though the evidence is still weak, this concise review provides some evidence of the necessity and anticipated results of performing clinical remount on existing dentures.

In this review, not only articles written in English but also those in Japanese were inspected. The Japanese Universal Health Insurance Coverage System (JUHICS) has been provided to all citizens in Japan since 1961 and was rated as one of the most effective insurance systems in the world [36,37]. JUHICS provides financial support for insured patients with new complete dentures every six months [38]. In addition, the Japanese have the longest life expectancy all over the world [39]. With the aging population geometry and an effective insurance system, the authors considered there should be some valuable studies in Japanese. Therefore, one observational study written in Japanese was reviewed.

The drawback of this concise review was the quality of the included articles. None of the included articles was quality RCT, and the evaluation methods applied were uneven. Considering the lack of studies in this particular field, the drawback is understandable, and more studies are requested. 

Hence, with the limits of knowledge, the authors would like to propose a hypothetical model to explain the treatment mechanism, possible indications and contraindications for performing clinical remount on existing dentures. The authors humbly hope this hypothetical model may provide some useful information for future research.

## 4. Hypothetical Model

How do existing prostheses benefit from the clinical remount procedure? According to the authors’ hypothetical model, the clinical remount procedure effectively and accurately corrects the accumulative errors of the jaw relationship.

### 4.1. Correction of the Accumulative Errors of Jaw Relationship

The first component of this model is ”correcting the accumulative errors of the jaw relationship.” Even when the prostheses were carefully balanced before delivery, many researchers have noted that discrepancies and malocclusion occur after the delivery of dentures. The reasons for the occlusal alterations can be classified into three categories: material factors, physiological changes and clinical practice. 

For the material factors, the acrylic denture base is porous, and mild distortion will occur due to absorption of water [21] (p. 159). Additionally, it is not uncommon to find worn occlusal surfaces on those dentures that had already served for a while [6,35]. For the physiological changes, factors including settling of the dentures [21] (p. 159), neuromuscular adjustment [14], edentulous ridge resorption [32,35] and denture wearing habits [32] were reported in previous studies. Last but not least, occlusal alterations will also occur due to adjustments carried out by clinicians in recall appointments.

Long-term cephalometric evaluation also shows that complete dentures exhibit a counterclockwise rotation and forward movement mainly because of soft tissue seating [40] or efforts made by patients to retain poorly fitting, worn-out prostheses [41].

The authors consider that the clinical remount procedure should be understood as an extraoral method to establish occlusion and articulation of the prostheses at any designated position according to the bite registration; however, if the original jaw relationship is difficult to duplicate [6] or is unstable, it is safer to re-establish the articulation of the used prostheses as close to the centric relation as possible.

Centric relation is widely considered to be the starting point of mandibular movement with reproducibility. From this position, the mandible can perform lateral and anterior eccentric movements [42,43]. Articulating the prostheses as close to the centric relation as possible maximizes the possibility that the mandibular movement coincides with the set articulation of the prostheses. This strategy also reduces the chance of horizontal deviation because the lateral movement border of the mandible converges gradually while retruding. Thus, the prostheses are more likely to be stabilized during the masticatory cycle (Figure 2) [44,45].

### 4.2. Selective Grinding on the Articulator

The second component of this model is “performing selective grinding on a stable platform.” Although dozens of papers promote the clinical remount procedure, almost all of them focus on the necessity and value of this procedure when inserting new dentures. Compared with the process of adjusting the occlusion in the patient’s mouth directly, the clinical remount procedure provides a solid, clean, saliva-free and easily accessible working environment. This procedure also reduces patient participation and alleviates psychological pressure [1] (p. 397).

Although the thickness of the articulating paper may vary from 25 to 100 μm, the physiological displacement of the edentulous ridge has been reported to be larger than 500 μm [3,4]. Research has indicated that compared with evaluating the occlusion of dentures on an articulator, marking the occlusal contact points with articulating paper in the patient’s mouth directly creates more spurious and false marks, regardless of whether the dentures have been stabilized [5,6]. Furthermore, it is difficult to check the occlusal status of each pair of molars individually intraorally. Mpungose et al. [7] established the low accuracy and poor reliability of evaluating the occlusal status by visual inspection only. The false, unclear marks lead to unnecessary grinding of the artificial teeth, decreasing the longevity of the dentures.

Additionally, edentulous patients today have different characteristics than those of earlier times, including greater mean age, more ridge resorption and increased loss of oral awareness and dexterity. These negative factors make an accurate and repeatable jaw relationship difficult to achieve. Last but not least, protracted treatment time and intimate recalls may be unfavored for the edentulous patients nowadays because of greater age and impaired general health [46,47].

Achieving the same extent of occlusal adjustment as the clinical remount procedure at the chairside directly would involve a challenging and lengthy appointment. From the patient’s perspective, the use of the clinical remount procedure means that chairside procedures can be shortened, and the repeated procedure of inserting and retrieving dentures between each grinding can be avoided. The reduction in patient participation also grants the clinical remount procedure more versatility.

Some researchers have raised doubts about the necessity of the clinical remount procedure, particularly those focused on simplifying the fabrication of new complete dentures [47,48,49,50,51,52]. With carefully determined vertical and horizontal jaw relationships, newly fabricated complete dentures seem to serve patients well even without the clinical remount procedure. Nevertheless, as previously discussed, an accumulation of errors in the jaw relationship seems inevitable. Because of the unknown extent of errors in the jaw relationship of dentures already in use, the authors believe that the clinical remount procedure should be undertaken to re-establish the occlusion and articulation if indicated.

Other researchers may doubt that an articulator can exactly duplicate mandibular movement. Discrepancies cannot be denied, and minor refinements may be needed even if the remounting procedure is carefully performed. However, as mentioned previously, the use of articulators reduces the number of chairside adjustments required [53]. Additionally, the authors believe that the repeatable eccentric movements of an articulator and the solid working environment make it easier to establish a credible broad zone of bilateral balanced articulation [42]. As Heartwell and Rahn claimed, the remaining discrepancies are so negligible that the resilience of the supporting tissues accommodates for the error [1] (p. 399). In our view, the resilience of the supporting tissue in conventional complete denture treatment is a double-edged sword. Although the resilience of tissues hinders accurate adjustments from being carried out intraorally, this property also helps patients cope with the minor occlusal interferences of their dentures. One of the goals of clinical remounting is to minimize occlusal discrepancies and make those discrepancies minor enough for the supporting tissue to compensate.

## 5. Indications and Contraindications

Performing a clinical remount procedure on existing dentures will effectively and accurately correct the accumulative errors of the jaw relationship. However, there is no clear treatment guideline about the indications and contraindications of this technique for existing dentures. Because of the limited information available, the authors offer the following suggestions for possible indications and contraindications.

### 5.1. Indications

The first indication is that the complaint is caused by occlusal discrepancies. Dentures with good retention but ones that become loose after conversation or chewing are likely to have faulty occlusion and articulation. Patients seldom complain precisely about the occlusion of their dentures, but they may complain about food being trapped between the ridge and the denture or uncertain sore spots without ulceration. As Heartwell and Rahn stated, one must assume that there are occlusal faults in all complete dentures until proven otherwise [1] (p. 394).

The second indication is poor masticatory function. Some patients have no complaints about their dentures because they have already given up chewing with their dentures. Patients may change their diet or swallow larger boluses of food to cope with decreased masticatory function [54,55,56]. Functional impairment may also not be recognized by patients. Masticatory function was previously evaluated with a reliable test food, such as carrots or peanuts, with the sieve method [52]. However, the complicated procedure of the sieve method is difficult to carry out in daily practice. Recently, a simplified but similar gummy testing kit has become commercially available in Japan and has been formally adopted as a diagnostic tool for oral hypofunction by the JUHICS [57]. Verifying masticatory function with the gummy testing kit or other standardized foods, such as a certain brand of cookie, may help clinicians gather objective information about masticatory function. If a patient is diagnosed with poor masticatory function, performing the clinical remount procedure on the existing dentures may serve as a good starting point for the rehabilitation of oral function.

### 5.2. Contraindications

The treatment efficiency of the clinical remount procedure may be limited if either of two following clinical findings is noted: an extreme anterior–posterior discrepancy between the maximal intercuspal position (MIP) and the centric relation (CR) or an extremely low vertical dimension.

An extreme amount of anterior–posterior discrepancy between MIP and CR will hinder the treatment efficiency of the clinical remount procedure. If the occlusion of the denture is re-established in CR, as previously described, the overlapping area of the maxillary and mandibular arches will be heavily restricted owing to the anatomical shape of the human dental arches. Adequate posterior support may be difficult to acquire in such a confined area; thus, the adjusted dentures cannot be well stabilized (Figure 3).

The second contraindication is extremely low occlusal vertical dimension. Removing occlusal discrepancies with selective grinding will decrease the vertical dimension slightly. If the existing dentures are already too thin to be ground, or the vertical dimension of the dentures is too low to achieve stable occlusal contacts intraorally, it may be more cost effective to make new dentures if there are no other special concerns (Figure 4).

## 6. Final Discussion

Although the hypothetical model, indications and contraindications serve well in the authors’ daily practice, this model is heavily clinician oriented. Thus, there are still several perspectives that lack discussion.

Performing clinical remount on dentures will inevitably lose some vertical dimension [34] (p. 271). Though the short-term results were reported by Schierano and Kawahara positively [32,33], the long-term effects of occlusal equilibrium and the loss of vertical dimension are still unknown. Despite the relationship between occlusion and TMDs still being controversial, several studies had reported that the incorrect vertical dimension and centric relation were the most frequent causes of temporomandibular disorders (TMDs) among denture wearers [58,59,60,61,62]. Further studies are needed to evaluate whether it is safe enough or not to acquire occlusal harmony with the mighty loss of vertical dimension in existing dentures.

In addition to TMDs, jaw movements were reported to have a close relationship with cervical muscles, even the body balance [63,64]. To the authors’ limited knowledge, though there are studies about the influences of dental occlusion on body balance [63], studies about denture occlusion and body balance are still scarce. The research of long-term treatment effects of performing clinical remount on existing dentures is requested.

This model should also be verified periodically with the involvement of new technologies and the changes in patients’ profiles in the future. Ninety years ago, Neil documented that “When cases are mounted upon a precision instrument and balanced, this balance will be fairly well maintained for a period of one year, at which time the occlusion should be re-balanced.” He also proposed patients should have new dentures at the end of the fifth year due to periodically clinical remounts leading to a severe loss of the vertical dimension eventually [20]. Nearly fifty years ago, Lauritzen considered patients should be recalled for “Denture Service” as soon as “they sense the first looseness of the dentures” or every 2–3 years [21].

Nowadays, new technologies, including CAD/CAM, 3D printing, kinematic facebows, improvements of materials and so on, will make making new dentures more precise and less prone to wear [65,66,67,68]. However, as aforementioned, the material factor is just one of the reasons that cause the alterations in occlusion. Moreover, even with the most advanced technologies, some hand fitting may still be needed from time to time. Therefore, instead of performing clinical remounts periodically, the authors tend to evaluate patients with the indications listed above.

Further prospective research and systematic reviews are needed to verify the validity of this treatment philosophy. As Zarb and Hobkirk stated, when fabricating new complete dentures, the need to remount will diminish with developing skill, but it will never go away [69] (p. 242). The authors believe that in our super-aging society, the clinical remount procedure is still an essential skill both when fabricating quality dentures and when maintaining those already in use.

## 7. Conclusions

This narrative literature review and hypothetical model portray a possible mechanism for performing the clinical remount procedure on existing complete dentures. The accumulated errors of the jaw relationship can be corrected effectively and accurately in a stable working environment. The clinical remount procedure is suitable for correcting the occlusion of existing dentures and should be performed if a patient has poor masticatory function or occlusion-related complaints.

## Figures and Tables

**Figure 1 healthcare-10-01067-f001:**
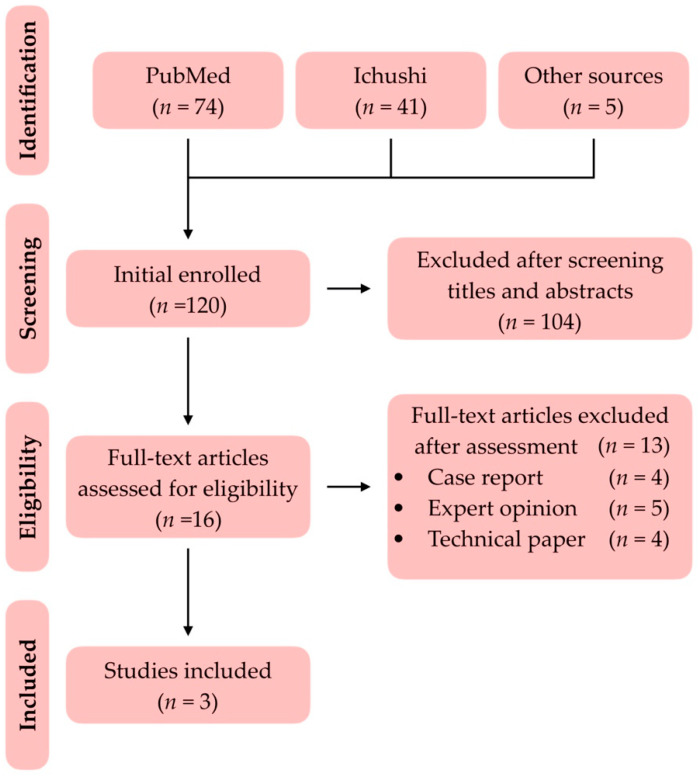
The diagram of the inclusion process.

**Figure 2 healthcare-10-01067-f002:**
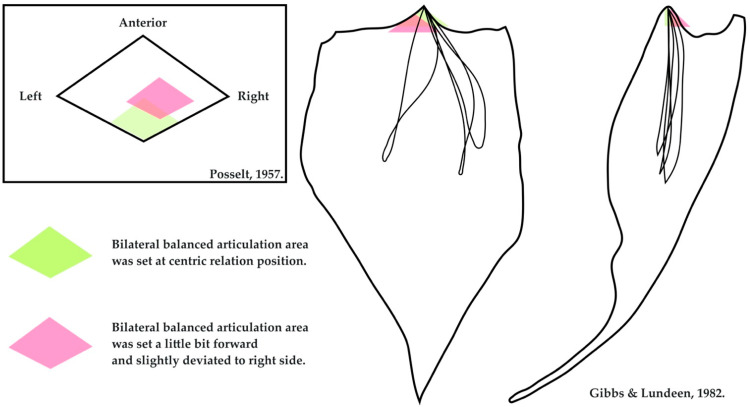
Diagram showing the advantage of setting the occlusion and articulation of dentures in the centric relation. The original figures come from Posselt and Gibbs and Lundeen [44,45].

**Figure 3 healthcare-10-01067-f003:**
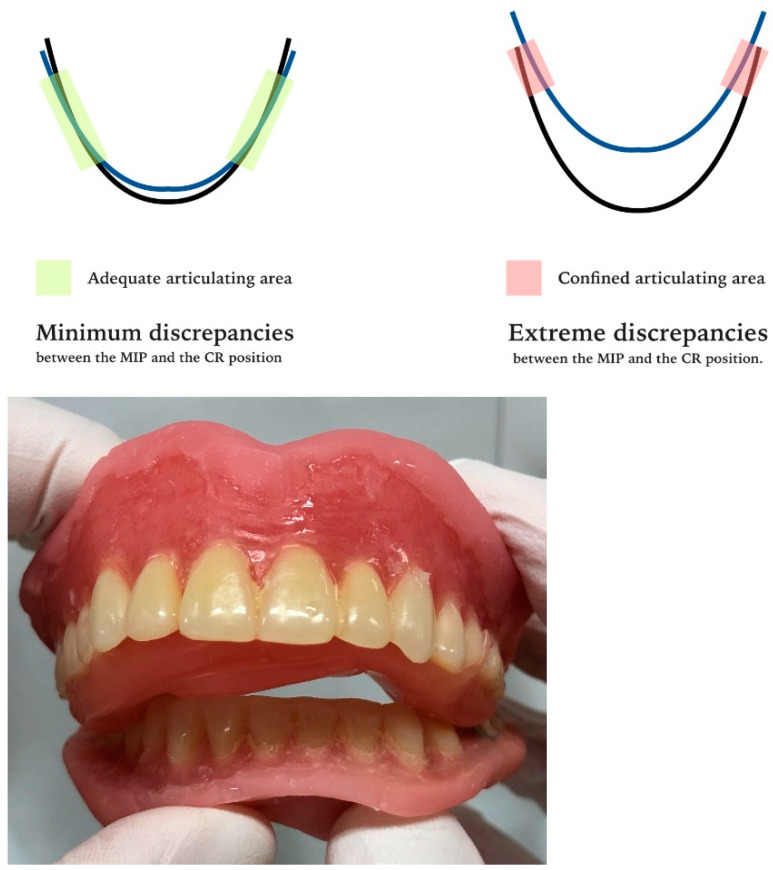
A case in which the clinical remount procedure is contraindicated on existing dentures. The extreme anterior–posterior discrepancy between the maximal intercuspal position (MIP) and the centric relation (CR) had reduced the posterior occlusal supporting areas, making it more difficult to stabilize the adjusted dentures.

**Figure 4 healthcare-10-01067-f004:**
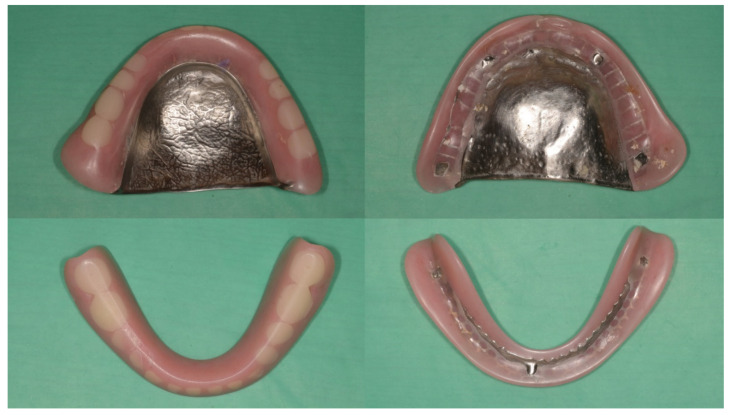
Another case in which the clinical remount procedure is contraindicated on existing dentures. The vertical dimension of the dentures is extreme low, and the artificial teeth are heavily worn.

**Table 1 healthcare-10-01067-t001:** The search methodology applied for each database.

Online Database	Dates of Coverage	Keywords Used
PubMed (NLM)	1977.04 to 2022.04	Clinical remount
Ichushi (Japan Medical Abstracts Society)	1983.03 to 2021.05	リマウント (Rimaunto)

**Table 2 healthcare-10-01067-t002:** The excluded articles [13,15,20,21,23,24,25,26,27,28,29,30,31].

Reason of Exclusion	Authors and Published Year	
Case report	Yamakage S	2001 [23]
	Kawahara S	2015 [24]
	Takamori A	2016 [25]
	Osada K	2020 [26]
Expert opinion	Neil E	1932 [20]
	Lauritzen AG	1974 [21]
	Gutowski A	1990 [13]
	Badel T	2001 [15]
	Uchiyama Y	2008 [27]
Technical paper	Hochstedler JL	1995 [28]
	Oh WS	2009 [29]
	Liu FC	2010 [30]
	Chauhan MD	2012 [31]

**Table 3 healthcare-10-01067-t003:** The current research works about performing clinical remount on existing dentures were listed and compared. (CR: Centric relation; CD: Complete denture; IOD: Implant-supported overdenture) [5,32,33].

Author(s)	Objective(s)	Mounting Procedures	Method of Evaluations	Treatment Group	Mean Age	Time Since Rehabilitation	Results and Discussions
Schierano, G. et al., 2002 [32]	To evaluate the influence of selective grinding on the thickness discrimination ability in patients wearing complete dentures.	Dentures were mounted at CR with facebow transfer.	Patients were instructed to bite the metal foil on bilateral premolar areas. The thickness of the foil was gradually increased until patients were perceived.	Conventional bimaxillary CD. (*n* = 12) (Group 1)	65 ± 1.543	4 ± 0.42 years	After selective grinding, the average thickness thresholds were decreased from 110μm to 90.5 μm in Group 1 and from 64μm to 37.67 μm in Group 2. The occlusal alterations affected nearly all the patients and were not correlated with the time since rehabilitation.
				Maxillary CD with mandibular IOD. (*n* = 12) (Group 2)	67 ± 1.543	4.42 ± 0.79 years	
Kawahara, H. et al., 2016 [33]	To evaluate masticatory function before and after the clinical remount procedure.	Dentures were mounted at CR without facebow transfer.	Patients were instructed to chew standardized gummy for 20 seconds and rinsed with 10 ml of water. The solution was retrieved and the concentration of the glucose was measured.	Single arch or bimaxillary CD. (*n* = 70)	75.8	Not mentioned.	The average reading was significantly increased from 135.1 to 184.4 mg/dL, indicating the increment of masticatory function. The participants with lower masticatory function would benefit most after the clinical remount procedure.
Atashrazm, P. et al., 2009 [5]	To investigate the prevalence of occlusal disharmony in inserted complete dentures and its associated causes.	Dentures were mounted at CR with facebow transfer.	The occlusion of dentures was checked with 60μm articulating by tactile sensation. Occlusal disharmony was defined as an absence of simultaneous bilateral contacts at CR.	Conventional bimaxillary CD. (*n* = 107)	57	30 days or less after insertion.	31 patients showed occlusal disharmony. 25 of them had no clinical remount procedure done at the insertion appointment. An odds ratio of >35 showed the probability of occlusal disharmony occurring in CDs with no clinical remount performed is 35 times greater than those remounted.

## Data Availability

Not applicable.

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
