# Peer review of "Hypothetical Model of How a Clinical Remount Procedure Benefits Patients with Existing Dentures: A Narrative Literature Review"

_healthcare, 2022, doi:10.3390/healthcare10061067_

Round 1

Reviewer 1 Report

This manuscript provides important and detailed data in a structured form of descriptive text and practical illustration (figures). There are only a few minor suggestions to consider for possible spelling/grammar improvement:

  • Line 22 and 244: “occlusal-related complaints” – more correct would be to use “-related” with a noun – “occlusion-related”
  • Line 96: a question mark is missing
  • Line 145-147: the sentence needs paraphrasing in order to achieve a better understanding
  • Line 149: a comma should be placed after “directly” in order to achieve a better understanding
  • Line 154: “and accuracy” is already crossed out – this needs to be erased
  • Line 240: a spelling error “compete” – should be “complete”

The paper is well written and can be published with a minor revision.

Author Response

Dear Reviewer 1:

Thank you for your positive feedback and pointing out the flaws of our article. We had modified the manuscript according to your comment. These details included:

  1. Line 22 and 244: “Occlusion-related”.
    We have replaced the word “occlusal-related” with “occlusion-related” in Line 22 and 244.

  2. Line 96: Missing of question mark.

We tried to use emphatic sentence but made the sentence difficult to understand accidently. Therefore, the sentence was modified with more plain description for easy understanding.
“According to the authors’ hypothetical model, the clinical remount procedure effectively and accurately correct the accumulative errors of the jaw relationship.”

  1. Line 145 - 147: Paraphrasing for easy understanding.

The sentence was modified as followed:
“Last but not least, protracted treatment time and intimate recalls may be unfavorited for the edentulous patients nowadays because of greater age and impaired general health”.

  1. Line 149: A comma was added. Thank you for your warm advice!

  1. Line 154: “And accuracy” was deleted as it should. Thank you for your pointing out!

  1. Line 240: The misspelling was corrected.

Once again, we appreciate your patient reading and delicate check for our manuscript.

Sincerely,

Ikiru Atsuta

Reviewer 2 Report

Please add a Table with the relevant details from the published articles.

Author Response

Dear Reviewer 2:

Thank you for pointing out the flaws of our article. We had modified the manuscript according to your comment. A table was added for comparing the published articles in this specific field.

Table 1. The current researches about performing clinical remount on existing dentures were listed and compared. (CR:Centric Relation; CD: Complete denture; IOD: Implant-supported overdenture.)

Once again, we appreciate your patient reading and delicate check for our manuscript.

Sincerely,

Ikiru Atsuta

Reviewer 3 Report

Dear Sirs,

unfortunately, to my mind this paper does not combine to any of the known standards. This is a perfect theme for metaanalysis or systematic review and only in this form could be accepted for the further steps of the review.

In lines 100-107, the Authors should pay attention to rotation of the mandible - in this case the autorotation of the mandible could be related to the set of soft tissues or tooth wear. The lower clearance gap, the higher rotation of the mandible. The Authors should reevaluate those findings.

There is also no information on the performance of the dentures. If the dentures were performed without previous relaxation, the chewing habits will remain and may cause the masticatory premature contacts. This is also a huge limitation of this study. 

You should also review English of this paper, as it is full of repetitions, even within one sentence.

My advice to you is rediscuss this topic with experienced specialist in TMD and orthodontics and reevaluate this paper to more complex - systematic review or mataanalysis.

Thank you and good luck.

Author Response

Dear Reviewer 3:

Thank you for your patient reading and delicate check for our manuscript. We had carefully discussed your valuable advice and made a difficult decision. We request for maintaining the major frame of this manuscript for several reasons:

  1. The difficulties of conducting a systematic review.

  In 2019, Verhaeghe et al published a systematic review of the clinical remount on complete dentures. However, the systematic review focused on new dentures, and only 4 studies were involved under strict review criteria.

  The narrative review presented aims to understand the potential benefits of performing clinical remount procedures on “existing dentures”. In the authors’ limited knowledge, currently, there are only 2 original studies in this field with different aims and experiment designs, making conducting a proper systematic review difficult to achieve. Surely, we commit to carry out quality systemic review or meta-analysis once there are enough published articles in this particular field.

  1. The difficulties of remaining clarity and engagement in a concise manuscript.

  We agreed on those details you pointed out, including the rotation of the mandible due to the decrement of vertical dimension, and possible premature contact may occur if patients maintain unfavorited chewing habits. However, the concerns you had were also well accommodated in our hypothetical model. Accumulative errors in jaw relationship need to be corrected to make denture function properly, and this goal can be achieved precisely and effectively with the clinical remount procedure. We tried keeping the manuscript well engaged and concise and we are sorry that the arrangement of tailored materials did not fulfill your requirements.

Once again, we appreciated your quality time and comments. We wish the explanations above may gain some consensus between us and further discussions are always welcomed.

Sincerely,

Ikiru Atsuta

Reviewer 4 Report

While this paper has no actual experimental or research data to contribute to the topic, it is nonetheless a very useful study in relation to the problem of dental occlusion, and management thereof. The literature review provides a good foundation for the presentation of the hypothesis. In my opinion, this is a very clinically useful study for active practitioners, and provides an excellent foundation for future research. I have no concerns with the content of this manuscript as such. It is well written, and the hypothesis well argued.

Author Response

Dear Reviewer 4:

Thank you for your patient reading and delicate check for our manuscript. This research field is still relatively new, and more research is still needed. We commit to conducting further studies and hope our results bring some positive effects in this field.

Sincerely,

Ikiru Atsuta

Round 2

Reviewer 3 Report

Dear Authors,

We request for maintaining the major frame of this manuscript for several reasons:

 Although I totally disagree with you in that point, here are my suggestions how to make this work look more like "according to the last standards" with respect of non-conducting a systematic review. I accept your explanations, although still my personal oppinion would be that metaanalysis would be the best way to present that topic. Anyway, please combine with my suggestions:

1. Please, prepare an illustration on how the paper was written, eg. 1 researcher going through the databases, 2nd researcher conducting the critical review, etc. - this should be a "sine qua non condition" of this paper

  1. In 2019, Verhaeghe et al published a systematic review of the clinical remount on complete dentures. However, the systematic review
  2. focused on new dentures, and only 4 studies were involved under strict review criteria.

In this situation, the Authors should incorporate this valid paper to the discussion

  1. The difficulties of remaining clarity and engagement in a concise manuscript.

  We agreed on those details you pointed out, including the rotation of the mandible due to the decrement of vertical dimension, and possible premature contact may occur if patients maintain unfavorited chewing habits. However, the concerns you had were also well accommodated in our hypothetical model. Accumulative errors in jaw relationship need to be corrected to make denture function properly, and this goal can be achieved precisely and effectively with the clinical remount procedure. We tried keeping the manuscript well engaged and concise and we are sorry that the arrangement of tailored materials did not fulfill your requirements.

 This should also be incorporated in the discussion section. 

In the discussion section, there should be several more topics mentioned, giving a perspective to manufacture better, less prone to tooth wear dentures for the future (maybe you should call this "perspectives"):

1. Incorporation of CAD/CAM procedures and resigns in prefabrication of the dentures:

Raszewski Z. Acrylic resins in the CAD/CAM technology: A systematic literature review. Dent Med Probl. 2020;57(4):449–454. doi:10.17219/dmp/124697

2. Incorporation of use of facebows and novel techniques:

- Wieckiewicz M, Zietek M, Nowakowska D, Wieckiewicz W. Comparison of selected kinematic facebows applied to mandibular tracing. Biomed Res Int. 2014;2014:818694. doi: 10.1155/2014/818694. Epub 2014 May 7. PMID: 24895613; PMCID: PMC4033495.

Caruso S, Storti E, Nota A, Ehsani S, Gatto R. Temporomandibular Joint Anatomy Assessed by CBCT Images. Biomed Res Int. 2017;2017:2916953. doi: 10.1155/2017/2916953. Epub 2017 Feb 2. PMID: 28261607;

Green JI. Prevention and Management of Tooth Wear: The Role of Dental Technology. Prim Dent J. 2016 Aug 1;5(3):30-33. doi: 10.1177/205016841600500302. PMID: 28826461.

Goiato MC, Garcia AR, dos Santos DM, Pesqueira AA. TMJ vibrations in asymptomatic patients using old and new complete dentures. J Prosthodont. 2010 Aug;19(6):438-42. doi: 10.1111/j.1532-849X.2010.00614.x. Epub 2010 Jun 8. PMID: 20546491.

- refer to MODJAW as well 

3. The discussion with how the model of chewing changes within the years:

Österlund C, Nilsson E, Hellström F, Häger CK, Häggman-Henrikson B. Jaw-neck movement integration in 6-year-old children differs from that of adults. J Oral Rehabil. 2020 Jan;47(1):27-35. doi: 10.1111/joor.12865. Epub 2019 Aug 23. PMID: 31357241.

4. It should also be widened by an aspect of good or bad retention

Jassim TK, Kareem AE, Alloaibi MA. In vivo evaluation of the impact of various border molding materials and techniques on the retention of complete maxillary dentures. Dent Med Probl. 2020 Apr-Jun;57(2):191-196. doi: 10.17219/dmp/115104

The limitations of the study should be a separate chapter in this research, including the limitations on the fabrication of the dentures "history" and other aspects mentioned by us in the Reviewer-Author discussion.

I think those aspects will make you become closer to the novel standards of scientiffic reports, not changing too much the manuscript body.

Best regards and good luck 

Author Response

The modification report for Reviewer 3

Dear Reviewer 3:

Thank you for pointing out the flaws of our article once again. We appreciate your time and valuable advice. After reviewing the articles you provided and some additional references, we have heavily modified the manuscript, and hopefully, reach your standard as close as possible.

These modifications included:

  1. Material and methods were added, including Table 1, 2, and Figure 1.
  2. One more article was reviewed (Chapter 3.2).
    According to our updated workflow, one more observational research (Atashrazm P, 2009) was found and included in our review.
  3. A brief discussion (Chapter 3.3) and the limitations of this concise review were added.
  4. The final discussion (Chapter 6) was added. Thanks to your broad perspectives, our manuscript became more comprehensive and the future research plan was also clarified by our team.

Although it was a little bit harsh for adding these major modifications in a short period, we cannot thank you enough for providing clear, specific, and experienced advice. Thank you for making this manuscript better than our expectations. It means a lot to us. We are confident with our modifications and hope you enjoy it as well! Have a nice day!

Sincerely,

Chi-Hsiang Cheng & Ikiru Atsuta
